# Cytotoxic Action of Artemisinin and Scopoletin on Planktonic Forms and on Biofilms of *Candida* Species

**DOI:** 10.3390/molecules25030476

**Published:** 2020-01-22

**Authors:** Sourav Das, Lilla Czuni, Viktória Báló, Gábor Papp, Zoltán Gazdag, Nóra Papp, Tamás Kőszegi

**Affiliations:** 1Department of Laboratory Medicine, University of Pécs, Medical School, 7624 Pécs, Ifjúság u. 13., Hungary; pharma.souravdas@gmail.com; 2János Szentágothai Research Center, University of Pécs, 7624 Pécs, Ifjúság u. 20., Hungary; 3Department of General and Environmental Microbiology, Institute of Biology, University of Pécs, 7624 Pécs, Ifjúság u. 6., Hungary; czuni.lilla@gmail.com (L.C.); balo.viktoria@gmail.com (V.B.); pappgab@gamma.ttk.pte.hu (G.P.); gazdag@gamma.ttk.pte.hu (Z.G.); 4Microbial Biotechnology Research Group, János Szentágothai Research Center, University of Pécs, 7624 Pécs, Ifjúság u. 20., Hungary; 5Department of Pharmacognosy, University of Pécs, Faculty of Pharmacy, 7624 Pécs, Rókus u. 2, Hungary

**Keywords:** *Candida* species, artemisinin, scopoletin, oxidative stress, mature biofilm, antifungal effect

## Abstract

We investigated the antifungal activities of purified plant metabolites artemisinin (Ar) and scopoletin (Sc) including inhibition, effects on metabolic activities, viability, and oxidative stress on planktonic forms and on preformed biofilms of seven *Candida* species. The characteristic minimum inhibitory concentration (MIC_90_) of Ar and Sc against *Candida* species ranged from 21.83–142.1 µg/mL and 67.22–119.4 µg/mL, respectively. Drug concentrations causing ≈10% CFU decrease within 60 min of treatments were also determined (minimum effective concentration, MEC_10_) using 100-fold higher CFUs than in the case of MIC_90_ studies. Cytotoxic effects on planktonic and on mature biofilms of *Candida* species at MEC_10_ concentrations were further evaluated with fluorescent live/dead discrimination techniques. *Candida glabrata, Candida guilliermondii*, and *Candida parapsilosis* were the species most sensitive to Ar and Sc. Ar and Sc were also found to promote the accumulation of intracellular reactive oxygen species (ROS) by increasing oxidative stress at their respective MEC_10_ concentrations against the tested planktonic *Candida* species. Ar and Sc possess dose-dependent antifungal action but the underlying mechanism type (fungistatic and fungicidal) is not clear yet. Our data suggest that Ar and Sc found in herbal plants might have potential usage in the fight against *Candida* biofilms.

## 1. Introduction

Fungal biofilms play an important role in numerous infections. They are composed of structural microbial communities adhering to different surfaces and being enveloped by the exopolymeric matrix [1]. Development of biofilms hinders the action of the defense system of the host and increases the resistance to standard antifungal agents [2]. *Candida* species like *Candida albicans (C. albicans*)*, Candida dubliniensis* (*C. dubliniensis*)*, Candida tropicalis* (*C. tropicalis*)*, Candida krusei* (*C. krusei*)*, Candida glabrata* (*C. glabrata), Candida guilliermondii* (*C. gulliermondii*), and *Candida parapsilosis* (*C. parapsilosis*) are fungal species of major medical importance. *Candida* species, members of human microflora, are diploid polymorphic yeasts [3] that can be found in the human gastrointestinal, respiratory, and inside the genitourinary tracts; therefore, *C. albicans* is found in vaginal, mucosal, and deep tissue infections [2,4,5]. In certain individuals with immune-compromised status, the *Candida* species might induce consequent infection. Also, *Candida* species can rapidly adapt to the host’s micro-environmental circumstances caused by pH and nutritional changes in the gastrointestinal tract [1]. Environmental imbalance because of nutritional change or pH shifting facilitates the abnormal growth of *Candida* species which results in candidiasis [6]. *Candida* species attack the gut epithelium barriers to reach the bloodstream via micro-fold cells (M cells) that are found in the gut-associated lymphoid tissue (GALT) of Peyer’s patches in the small intestine promoting intestinal infections [4]. Esophageal candidiasis is one of the major and common infections in people living with HIV/AIDS [1]. Symptoms of candidiasis in the mouth, throat, and esophagus generally include swallowing problems and pain [7,8]. Because of the increasing incidence of candidiasis and the difficulties in its treatment because of the limited options in the use of antifungal drugs with species-specific efficacy [9], there is an ultimate need for at least prevention of fungal infections. 

Artemisinin (Ar), belonging to a family of sesquiterpene lactones originally derived from *Artemisia annua* L. is well-known for its anti-malarial actions by forming free radicals through cleavage of intra-parasitic iron-endoperoxide groups and by alkylation of specific malarial proteins mediating eradication of *Plasmodium* species [10]. Protection against cancers and inflammation by artemisinin has also been documented [11]. Ar has been reported to possess anti-infective activity against viruses (Human cytomegalovirus) and fungi (*Cryptococcus neoformans*) [12].

Scopoletin (Sc, 6-methoxy-7-hydroxycoumarin) is a phenolic compound isolated from several plants including *A. annua* L. [13]. Sc possesses anti-tumor and anti-angiogenesis properties by initiating cell cycle arrest and facilitating apoptosis in human prostate tumor cells and leukemia cell lines [14,15]. Although anti-fungal effects including anti-*Candida* activity of Ar and Sc were already studied, yet no detailed data are found in the literature on *Candida* species’ viability and biofilm formation treated with Ar and Sc at their minimum effective concentration (MEC_10_). Moreover, there are no experimental results obtained by using the double-stain rapid fluorescent assay for live/dead cellular discrimination along with metabolic activity determination for Ar- and Sc-exposed *Candida* species.

Because of the diverse biological activities including reactive oxygen species (ROS) generation [13,14,16,17], we hypothesized that both Ar and Sc having a wide range of biological actions might exert antimicrobial activity as well. Therefore, the aim of our present study is to contribute to the knowledge of the hypothesized anti-microbial and anti-biofilm activity of Ar and Sc regarding their effects on *C. albicans* and on non-*albicans* species. The effect of Ar and Sc on cellular viability, metabolic activity (enzymatic reducing ability) and oxidative stress balance of the *Candida* species was evaluated by SYBR green I-propidium iodide double staining method, resazurin assay, dihydroethidium (DHE) and dihydrorhodamine 123 (DHR 123) fluorescence assays, respectively. The biomass and metabolic activity of the living cells in the mature biofilms were assessed by colorimetric and double fluorescence staining techniques. 

## 2. Results

### 2.1. Antifungal Activities (MIC_90_) of Artemisinin and Scopoletin 

Average MIC_90_ data obtained after 48 h of treatment with Ar and Sc for seven *Candida* spp. are presented in Figure 1. Ar MIC_90_ ranged from 21.83 to 142.1 µg/mL, while Sc MIC_90_ was between 83.43 and 119 µg/mL taking all the tested *Candida* species. Ar MIC_90_ values were significantly lower than those of Sc in the case of *C. dubliniensis*, *C. krusei,* and *C. parapsilosis* (*p < 0.01*). *C. glabrata* and *C. tropicalis* was more susceptible to Sc (67.22 and 119 µg/mL, respectively). However, in general Ar expressed higher fungicidal activity than Sc at concentrations in the range of 27 to 80%, less than was seen for Sc. 

### 2.2. Data of Minimum Effective Concentrations (MEC_10_) for Planktonic Candida Species

Dose-response curves regarding the minimum effective concentrations (MEC_10_) for Ar and Sc on the selected opportunistic *Candida* species are shown in Figure 2. For Ar the obtained MEC_10_ concentrations are as follows: *C. albicans* 84.04 ± 2.92; *C. dubliniensis* 71.26 ± 3.81; *C. tropicalis* 89.95 ± 3.62; *C. krusei* 233.14 ± 4.72; *C. glabrata* 324.97 ± 4.42; *C. guilliermondii* 180.9 ± 3.36; and *C. parapsilosis* 171.86 ± 4.24 (in µg/mL). For Sc the MEC_10_ values were: *C. albicans* 132.17 ± 4.19; *C. dubliniensis* 132.17 ± 5.08; *C. tropicalis* 134.27 ± 4.66; *C. krusei* 162.01 ± 5.03; *C. glabrata* 215.38.97 ± 5.41; *C. guilliermondii* 149.2 ± 5.32, and *C. parapsilosis* 164.7 ± 5.26 (in µg/mL). The curves have expressed a dose-dependent cell survival (CFU count) after 60 min exposure to Ar and Sc. The doses corresponding to MEC_10_ (meaning an average 90% survival rate of mid-log phased populations, at ~10^5^ CFU/mL) were further used to evaluate their effects on planktonic cells and on mature biofilms. Cs, Ar, and Sc have shown an average MEC_10_ of 6.66 ± 0.14 µg/mL, 165.16 ± 3.87 µg/mL, and 160.84 ± 4.99 µg/mL, respectively for the tested planktonic *Candida* spp. Ar appeared to have the lowest MEC_10_ in the case of *C. dubliniensis* (21.83 ± 2.85 µg/mL), *C. krusei* (27.92 ± 2.55 µg/mL), and *C. parapsilosis* (27.98 ± 4.69 µg/mL), respectively.

### 2.3. Effects on Mature Biofilms

A variable response to Ar, Sc, and Cs at MEC_10_ doses have been found. Although no changes in the biomass have been observed in the case of Ar treatments, yet an average 55.04 ± 9.46% and 46.32 ± 12.62% reduction in metabolic activities and non-viable cells were seen in the case of *C. albicans*, *C. dubliniensis*, *C. tropicalis,* and *C. glabrata* when compared to growth control (*p < 0.01*). A significant reduction up to an average of 60% in the total biomass on Sc-treated *C. albicans*, *C. dubliniensis* and *C. glabrata* (*p < 0.01*) was found when compared with growth control (Gc) and Cs treatments. However, the data obtained from metabolic activity testing and non-viable cell numbers are invalid for the above species because of the significant biomass loss. *C. guilliermondii* has shown significantly higher resistance to Ar and Sc compared with the other species. The highest cell death (>60%) detected by the double fluorescence staining assay and the reduced metabolic activities (<50%) for *C. krusei*, and *C. parapsilosis* were found in the case of Sc treatment (Figure 3). 

### 2.4. Live/Dead Planktonic Cell Viability Discrimination

The long-term effects of Ar and Sc on the viability were tested on mid-log phased planktonic *Candida* spp. as well. Ar and Sc decreased the viability of the tested *Candida* spp. with an average viability reduction of 58.2 ± 3.5% and 58.6 ± 4.2% after 8 h, whereas 33.6 ± 3.9% and 32.2 ± 3.6% were seen after 16 h when compared with their respective controls. *C. glabrata, C. guilliermondii,* and *C. parapsilosis* were found to have less than 30% of viable cells in the presence of Ar and Sc for 16 h at their MEC_10_ concentrations. Ar and Sc showed a reduction in viability ≤50% after 8 h of treatment on *C. parapsilosis* and *C. krusei* compared with the controls (Figure 4). 

### 2.5. Effects on Metabolic Activity and on Colony Formation of Planktonic Cells

The metabolic activity of the mid-log phased planktonic *Candida* spp. in the presence of Ar and Sc at their MEC_10_ concentrations was evaluated with resazurin at 0, 8, and 16 h time points (Figure 5). Ar and Sc showed an average reduction in the metabolic activities of tested *Candida* spp. to two-fold after 8 h of treatment followed by three-fold after 16 h of treatment when compared with their respective controls. A less than two-fold decrease of metabolic activity was found in *C. glabrata, C. guilliermondii,* and *C. parapsilosis* planktonic cells in the presence of Ar and Sc after 16 h of treatment. 

A significant change in the planktonic cell number reduction compared with the control (*p < 0.01*) was also found after 8 and 16 h of treatment (Figure 6). A prominent reduction ≤50% of planktonic cell population, when compared to the controls, was found in the case of *C. glabrata, C. guilliermondii* and *C. parapsilosis* after 16 h of Ar, Sc, and caspofungin (Cs) exposures. 

### 2.6. Effects on Planktonics’ Oxidative Balance

The effects of Ar and Sc on the induction of the oxidative stress at MEC_10_ concentrations on respective mid-log phased *Candida* spp. planktonic cell populations are illustrated in Figure 7. Menadione (Me) was used as positive control. All the test drugs caused a significant oxidative stress in the planktonic cells compared with that of the controls (*p < 0.01*). More than 25% and 50% of O_2_^•−^ generation was observed in *C. glabrata, C. guilliermondii,* and *C. parapsilosis* compared with other tested *Candida* spp. after the treatment with Ar and Sc for 16 h. The overall O_2_^•−^ generation induced by Ar was found to be 13.4 ± 1.8% higher than that of Sc. Peroxide generation in *C. glabrata, C. guilliermondii,* and *C. parapsilosis* was also found to be higher (an average of 30.2% increment in Ar treatment and 8.4% increment in Sc treatment) compared with the other tested *Candida* spp. An average increment of 64.1 ± 2.6% was found in oxidative stress induction by Ar when compared to Sc in *Candida* species. In summary, we found that all the tested *Candida* species were susceptible to Ar and Sc treatments in favor of Ar vs. Sc. 

## 3. Discussion

Ar and Sc were effective against the selected *Candida* spp. and their antifungal activities were comparable in every measured parameter to Cs even if the antifungal agent’s MIC_90_ value was much lower than that of our plant-derived test compounds. The susceptibility of selected *Candida* spp. to Ar and Sc has not been reported before. The antifungal activity of Ar and Sc may be attributed to the inhibition of efflux pumps as it was shown for berberine, a natural isoquinoline alkaloid [18]. The measured lower effects of Sc is not known, however, the presence of efflux pump proteins belonging to ABC (ATP Binding Cassette) and MFS (Major Facilitators) superfamily might also be the responsible factors [18,19]. 

The resazurin assay and SYBR green I-propidium iodide double-stain fluorescent method enabled us to characterize both the planktonic cell populations and biofilms after the treatment with Ar and Sc. As a limitation of our viability study, it should be mentioned that measuring a single parameter only (e.g., the resazurin assay) does not necessarily reflect total cell viability because cellular ATP levels may change rapidly without significant reduction in intracellular enzyme activities [20]. Based on literature data both Ar and Sc may induce time-dependent cell wall and membrane damage enabling propidium iodide to bind to fungal nucleic acids [21]. The double-stain fluorescence assay showed that the longer the exposure time is, the higher cellular death rate is found in the Ar and Sc-treated fungal populations. The difference in viability among the examined *Candida* species we think is due to the potential difference in the activity of multidrug efflux pumps among the *Candida* spp. The maintenance of metabolic activities at about 50% even in the presence of low planktonic cell populations indicates the adaption of surviving planktonic cells [22]. 

Data presented in this study suggest the induction of apoptosis-like processes in the tested *Candida* spp. that may be due to the accumulation of reactive oxygen species (ROS) that induce or regulate the apoptosis in yeasts [23,24]. Using the superoxide and peroxide radicals’ fluorescent assays, we showed the accumulation of reactive oxygen species, which are the indicators of lipid damage [25]. It has been postulated that Ar and Sc affect ergosterol synthesis [19,26]. Ergosterol, apart from maintenance and regulation of structural and functional integrity of the membrane, inhibits lipid peroxidation [27]. This can also facilitate the permeability of the cell membrane and incorporation of propidium iodide through the compromised membranes into the cells. Further characterization of the effects of Ar and Sc on the levels of superoxide dismutase and catalase at increased ROS level must be assessed [26,28]. Moreover, Ar also plays an important role in the electron transport chain by overexpressing nuclear distribution protein nude homolog 1 (*NDE_1_*) that is responsible for encoding mitochondrial NADH dehydrogenases causing sensitivity to Ar followed by membrane disruption resulting in mitochondrial dysfunction by the higher free radical generation when compared to Sc [29,30]. 

In this study, we have further investigated the effects of Ar and Sc on the preformed mature biofilms of different *Candida* spp. Our results demonstrated that Ar is more effective in disrupting the preformed complexed, extracellular matrix-covered biofilm structure and killing the sessile (surface-attached) cell population as well compared to Sc at their respective MEC_10_ concentrations after 24 h exposure. This may be due to the abilities of the sesquiterpenoids to have action on amyloid proteins, which are one of the major building blocks of microbial biofilms [31,32,33,34]. On the other hand, Sc treatment showed a dramatic decrement in the total biomass including cell loss from the polystyrene surface in case of *C. albicans*, *C. dubliniensis,* and *C. glabrata,* which might be due to the action of coumarin derivatives on the chemical pathways such as quorum sensing resulting in biofilm dispersion [35,36,37,38,39,40]. However, the higher metabolic activities and the less cell death rate found in the case of *C. gulliermondii* indicates the presence of other drug resistance mechanism factors apart from poor drug penetration into *Candida* biofilms. The alternate mechanism might be related to the expression patterns of genes coding for multidrug efflux pumps [22,41,42]. Moreover, this might be the reason why the surviving cells can adapt to the stressful microenvironment by decreasing their metabolic activities [23,43]. Though the adapted cell populations with reduced metabolic activities might survive the toxicity of both test drugs, Ar exerts higher cell killing properties than those of Sc in the mature biofilm, whereas Sc might cause changes in the biofilm adherence factors to the polystyrene surfaces. Adaptation of *Candida* species under various conditions has already been described [23,44]. 

Based on the results of the tested *Candida* planktonic cells, we found that the MEC_10_ concentrations of our treating compounds are more effective to initiate oxidative imbalance and reducing metabolic activities followed by death of the planktonic cells when compared to the effects on preformed mature biofilms, which have shown variable results including cell death, biomass loss, and resistance. The reason behind the resistance to the MEC_10_ concentrations maybe due to the extracellular matrix acting as a diffusion barrier [39,45,46] or the up-regulations of the genes coding the efflux pumps [47,48,49]. Previous studies also reported that the planktonic cells released from the biofilms express subtle genetic changes for producing resistant phenotypes. As a consequence, the resistant new population may form biofilms with altered phenotype that might be responsible for the hostility in some of the tested *Candida* species [50].

The widely accepted crystal violet assay was used to investigate the effects of the tested compounds on the changes of *Candida* biofilm biomass [51]. This chemically basic dye stains fungal cells including negatively charged surface molecules and polysaccharides in the biofilm extracellular matrix enabling a rapid quantification of biofilm mass prior to labor-intensive microscope analysis [52]. Although, the poor correlation between the biofilm biomass reduction, metabolic activities and cellular viability remains a major limitation of this method because of the non-selective staining of the biofilm matrix including viable and dead cells as well [53]. Therefore, results obtained from the crystal violet assay must be combined with other multi-parametric techniques such as intracellular ATP, ATP/protein ratio, live/dead cell discrimination and selective visualization of the biomass matrix [54,55,56].

Our novel multi-parametric evaluations have more precisely highlighted the planktonic fungal susceptibility, killing ability inside the mature biofilm than the classical proliferation assays [57,58,59,60]. The observed antifungal effects and combined actions on mature biofilms of Ar and Sc might be useful for further detailed research for their potential usage as alternative antimicrobial agents for the treatment of *Candida* infections.

## 4. Materials and Methods

### 4.1. Materials

For our experiments, sterile 96-well microtiter plates for antifungal activity, live/dead discrimination and metabolic assays (Catalog number: 30096, SPL Life Sciences Co. Ltd., Gyeonggi-do, Korea), and for biofilm assays (Catalog number: 83.3924.500, Sarstedt AG & Co. KG, Numbrecht, Germany), crystal violet, peptone, yeast extract, agar-agar, potassium phosphate monobasic, acetic acid (Reanal Labor, Budapest, Hungary), resazurin, modified RPMI 1640 medium (containing 3.4% (w/v) MOPS, 1.8% (w/v) dextrose and 0.002% (w/v) adenine) [61], menadione (Me), artemisinin (Ar), scopoletin (Sc), SYBR green I 10,000x, propidium iodide, dihydrorhodamine 123 (DHR 123) and dihydroethidine (DHE) (Sigma-Aldrich Chemie GmbH, Steinheim, Germany), disodium phosphate, dimethyl sulfoxide (DMSO) from Chemolab Ltd. (Budapest, Hungary), sodium chloride (VWR Chemicals, Debrecen, Hungary), dextrose, adenine, potassium chloride (Scharlau Chemie S.A, Bercelona, Spain), 3-(N-morpholino) propanesulfonic acid (MOPS) (Serva Electrophoresis GmbH, Heidelberg, Germany), 0.22 µm vacuum filters from Merck Millipore (Molsheim, France) were used. All other chemicals in the study were of analytical or spectroscopic grade. Highly purified water (<1.0 µS) was applied throughout the experiments. Caspofungin (Cs) was purchased from Merck Sharp & Dohme Ltd. (Hertfordshire, UK).

### 4.2. Instruments Used in the Experiments

A microbiological incubator (Thermo Scientific Heraeus B12, Auro-Science Consulting Kft., Budapest, Hungary), microtiter plate reader (PerkinElmer EnSpire Multimode plate reader, Auro-Science Consulting Ltd., Budapest, Hungary), benchtop centrifuge (Hettich Rotina 420R, Auro-Science Consulting Kft., Budapest, Hungary) were used throughout the experiments.

### 4.3. Microorganisms and Culture Conditions

All the fungal species were obtained from Szeged Microbiology Collection, University of Szeged, Hungary (Table 1) and were maintained at the Department of General and Environmental Microbiology, Institute of Biology, University of Pécs, Hungary. *C. albicans* 1372, *C. dubliniensis* 1470, *C. tropicalis* 1368, *C. krusei* 779, *C. glabrata* 1374, *C. guilliermondii* 808, and *C. parapsilosis* 8006 were used to evaluate the MIC_90_, the minimum effective concentration (MEC_10_) for the planktonic cells and the inhibition of biofilm formation of the test samples. All the fungal species were cultured and maintained in yeast extract peptone dextrose agar medium (YPD: 1% (w/v) peptone, 0.5% (w/v) yeast extract, 2% (w/v) dextrose, 1.5% (w/v) agar-agar in distilled water [62]. Phosphate-buffered saline (PBS, pH 7.4) was from Life Technologies Ltd. (Budapest, Hungary), and highly purified water (<1.0 µS) was applied throughout the experiments.

### 4.4. Determination of Minimum Inhibitory Concentration (MIC_90_)

The MIC_90_ was performed according to a previously published method [63] with modifications. Ar and Sc were prepared in DMSO at concentrations ranging from 0.39–200 µg/mL and 0.01–8 µg/mL for Cs. Total of 100 µL of fungal suspensions of 10^3^ colony forming units (CFU)/mL in modified RPMI 1640 medium was pipetted into each well of sterile 96-well microtiter plates and mixed with 100 µL of dilutions of Ar and Sc. The final solvent concentration for the dilution of the drugs was restricted up to 2.5% v/v in the wells. Inoculated growth medium without any treatment was considered as the growth control. The sterile medium was taken as the blank. After 48 h of incubation at 30 °C in a microbiological incubator, the absorbance was measured at 595 nm. Absorbance values were converted to percentages compared with those of the growth control (≈100%) and data were fitted by non-linear dose-response curve method to calculate the dose producing ≥90% growth inhibition (MIC_90_). All the measurements were performed by applying three technical replicates in six independent experiments. Cs was used as a standard antifungal drug throughout the experiments.

### 4.5. Determination of Minimum Effective Concentration (MEC_10_)

The MEC_10_ measurement was performed according to our previously published protocol to estimate the dose-dependent survival rate of the cells [64,65,66] with modifications. Briefly, Ar and Sc were prepared in DMSO at concentrations ranging from 0.25–400 µg/mL and 0.5–32 µg/mL for Cs. Total of 100 µL of fungal suspensions of mid-log phased 10^5^ colony forming unit (CFU)/mL in modified RPMI 1640 medium were pipetted into each well of sterile 96-well microtiter plates and mixed with 100 µL of dilutions of Ar and Sc. The final solvent concentration for the dilutions of the drugs was restricted up to 2.5% (v/v) in the wells. Inoculated growth medium without any treatment was considered as the growth control. The sterile medium was taken as the blank. After 60 min of incubation at 30 °C in a microbiological incubator, 1 mL of treated and untreated samples was pipetted and were spread in nutrient agar media for 24 h at 30 °C for colony-forming unit (CFU/mL) quantification. CFU/mL values were converted to percentages and data were fitted with a non-linear dose-response curve to achieve drug concentrations producing an approximately 90% fungal cell growth (MEC_10_) compared to the untreated culture after one hour treatment. All the measurements were performed by applying three technical replicates in six independent experiments. Cs was used as a standard antifungal drug throughout the experiments.

### 4.6. Determination of the Effects on Preformed Mature Biofilms

The protocol for the measurement of the effects on the mature biofilms and treatment was adapted from the literature [7,67] with modifications. Total of 200 µL samples of 24 h old late-log phased *C. albicans* and non-*albicans* species in the modified RPMI 1640 medium (10^6^ CFU/mL) were used to culture biofilms without treatments for 24 h at 30 °C. The microtiter plates were washed carefully with sterile PBS (pH. 7.4) and were re-incubated with 200 µL culture medium containing Ar and Sc to be examined at MEC_10_ concentrations (µg/mL) for further 24 h at 30 °C. Modified RPMI media as blank, inoculated growth media as growth control (Gc) and Cs-treated samples were considered as controls throughout the experiments. The percentage (%) of inhibition was measured based on the comparison of the values with those of Gc. Treatments were performed with three technical replicates in six independent experiments.

#### 4.6.1. Evaluation of Total Fungal Biomass in the Biofilms

To determine the changes in the biofilms, a previously published crystal violet assay protocol was used [36] with modifications. After 24 h incubation, the crystal violet-stained and PBS-washed biofilms were treated with 200 µL of 33% (v/v) acetic acid in double-distilled water. Finally, after 20 min the acetic acid-dissolved dye from the biofilm matrix was pipetted into the wells of a new microtiter plate and the absorbance was measured at 590 nm. Cs at MEC_10_ concentration was used as positive control. The % biofilm biomass reduction measurement was based on growth control (Gc) values which were assigned to be 100% fungal biofilm mass. Six independent experiments were done with three technical replicates for each treatment.

#### 4.6.2. Resazurin-Derived Metabolism Assay in the Biofilms

A resazurin-based fluorescent assay published by Kerekes et al. was adapted [36] and was used to estimate the metabolically active (viable) cell number in the biofilm matrix treated with Ar and Sc at MEC_10_ concentrations. Briefly, after 24 h of treatment, the supernatants were removed and the wells were rinsed with PBS. Resazurin (12.5 µmol/L) in 200 µL of sterile PBS was added to each well containing the biofilm. After 40 min of incubation at 30 °C, the fluorescence was measured at excitation and emission wavelengths of 560/590 nm respectively. The % metabolic activity measurement was estimated based on the growth control (Gc) fluorescence values which were considered to be 100%. The PBS was taken as blank throughout the experiments. Six independent experiments were performed with three technical replicates for each treatment.

#### 4.6.3. Viability Assay of the Biofilms and of Planktonic Candida Species

For determination of live/dead microbes in the preformed mature biofilms we followed the method previously published [63]. Briefly, after 24 h of treatment with Ar, Sc, and Cs (positive control) at MEC_10_ concentrations in the wells of a microplate, the modified RPMI 1640 medium and the non-attached planktonic cells were removed, and the wells were rinsed and filled with 100 µL PBS. A working dye solution containing 20 µL SYBR green I (from 10,000× stock in DMSO, diluted 100 times in PBS) and 4 µL propidium iodide (500-fold dilution of working stock in DMSO prepared from 20 mmol/L in DMSO) in 1000 µL of PBS was used. A total of 100 µL of this working solution was added into the wells of the microplate. The plates were incubated at room temperature in the dark with mild shaking for 15 min.

For the evaluation of the long-term effects of Ar, Sc, and Cs at their MEC_10_ on the tested *Candida* spp., a previously published live/dead planktonic cell discrimination [63] was optimized and used. A wide concentration range of SYBR green I (5.5-5500-fold logarithmic dilutions) and propidium iodide (5-500-fold logarithmic dilutions) was examined. For the discrimination of live/dead cells, mid-log phased cell populations of 10^5^ CFU/mL were treated by Ar and Sc with a dose assigned for MEC_10_ concentrations and incubated at 30 °C for 16 h. Sampling was done at 0, 8, and 16 h time points for the measurements. The samples were centrifuged at 1000*g* for 5 min, washed in PBS and re-suspended in PBS (100 µL in each well). Then 100 µL of freshly prepared working dye solution in PBS (using 20 µL SYBR green I and 4 µL propidium iodide diluted solutions as described earlier) was added to the samples. The plate was incubated at room temperature for 15 min in the dark with mild shaking.

A PerkinElmer EnSpire multimode plate-reader was used to measure the fluorescence intensities of SYBR green I (excitation/emission wavelengths: 490/525 nm) and propidium iodide (excitation/emission wavelengths: 530/620 nm), respectively. A green to red fluorescence ratio for each sample and for each dose was achieved and the % of dead cells with the response to the applied dose was plotted against the applied Ar and Sc doses using the previously published formula [63]. In the case of biofilms, the measurements were done by area scan mode of the instrument. All treatments were done in triplicates and six independent experiments were performed.

### 4.7. Determination of the Metabolic Activity with Resazurin Assay and Colony Formation of Planktonic Fungal Cells

The metabolic activity of the cells under treatment conditions was performed by the widely accepted resazurin fluorescence method [68,69]. The experiments had to be optimized because the resazurin concentration, the cell number and the incubation time are crucial in order to increase the signal to noise ratio. Fluorescence data were corrected by subtracting background fluorescence (resazurin in PBS). The initial *Candida* cell density, resazurin levels, and duration of incubation time were varied by using resazurin concentrations ranging from 1.6 µmol/L to 25 µmol/L in the cell suspensions. The proper incubation time for resazurin exposure at an optimized concentration (12.5 µmol/L) was performed by 60 min incubation and sampling at intervals. The appropriate *Candida* species cell density was obtained by exposing a serial dilution of cells to 12.5 µmol/L resazurin in PBS. PBS without resazurin but with identical cell density to the investigated samples was used as blank.

For fungal colony forming unit measurements, we followed a previously published protocol [70]. Briefly, 1 mL of untreated and Ar, Sc, and Cs-treated sample was pipetted at 0, 8, and 16 h time intervals and was diluted 10^5^ times followed by spreading 50 μL onto 20 mL YPD agar medium using a cell spreader and incubated at 30 °C for 24 h. Cs was used as a reference control. Fungal colony forming units (CFU/mL) were determined, performed in triplicates and were plotted against time (h). Six independent experiments were done with three technical replicates for each treatment.

### 4.8. Detection of Peroxide (O_2_^2−^) and Superoxide Anion (O_2_^•−^) Generation in Planktonic Fungal Cells

We adapted the protocols previously described [25] for peroxide anion radicals (O_2_^2-^) determination and for superoxide detection in the cells [62] with modifications. The final cell population for the experiment was set to mid-log phased 10^5^ CFU/mL. An end concentration of 0.5 mmol/L of menadione (Me) in cell suspensions was added as the positive control. The cell suspensions were treated with MEC_10_ concentrations of test samples. The treatments were done for 60 min at 30 °C in the modified RPMI 1640 medium. Thereafter, the cells were centrifuged for 5 min at 1500*g* at room temperature. The pellets were re-suspended in PBS in the same volume. DHR 123 at a final concentration of 10 µmol/L for peroxides determination and DHE at a final concentration of 15 µmol/L for O_2_^•−^ determination were added separately to the cell samples. The stained cell samples were further incubated for 60 min at 30 °C in the dark condition with mild shaking. The samples were centrifuged and re-suspended in PBS followed by distribution of the samples into the wells of 96-well microplates. The fluorescence was measured at excitation/emission wavelengths of 500/536 nm for peroxides and 473/521 nm for O_2_^•−^ detection, respectively by a PerkinElmer EnSpire multimode plate-reader. The percentage increase in oxidative stress was measured by comparing the signals to those of the growth controls. Six independent experiments were done with three technical replicates for each treatment.

### 4.9. Statistical Analysis

All data are given as mean ± SD. Graphs and statistical analyses were conducted using OriginPro 2016 (OriginLab Corp., Northampton, MA, USA). All experiments were performed independently six times in triplicates and data were analyzed by one-way ANOVA test. *P*-value of <0.05 was considered as statistically significant. The minimum inhibitory concentration (MIC_90_) and the minimum effective concentration (MEC_10_) were calculated using a non-linear dose-response curve function as follows:(1)$$y=A_{1}+\;\frac{A_{2}-A_{1}}{1+10^{(LOG_{x}0-x)p}}$$
where, A_1_, A_2_, LOG_x_0, and p are the bottom asymptote, top asymptote, center and hill slope of the curve have been considered.

## Figures and Tables

**Figure 1 molecules-25-00476-f001:**
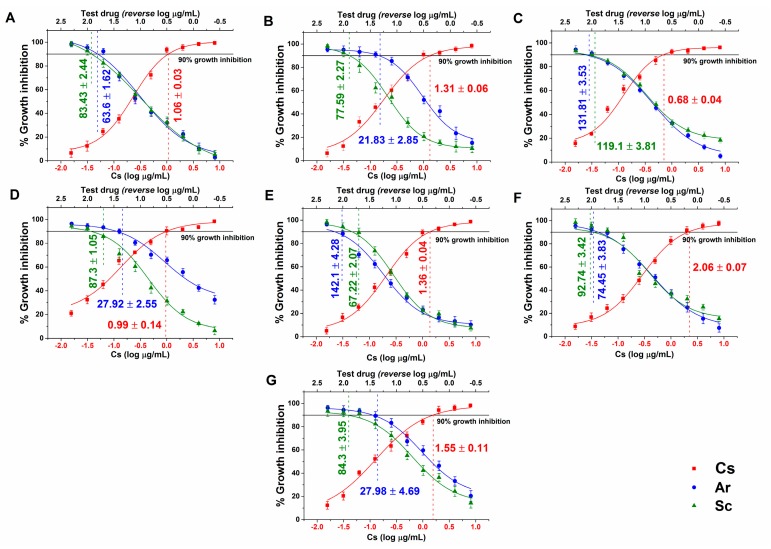
Minimum inhibitory concentrations (MIC_90_) (mean ± SD) of artemisinin (Ar) and scopoletin (Sc) on *C. albicans* (**A**), *C. dubliniensis* (**B**), *C. tropicalis* (**C**), *C. krusei* (**D**), *C. glabrata* (**E**), *C. guilliermondii* (**F**) and *C. parapsilosis* (**G**) species. Six independent experiments, each in three replicates, compared with caspofungin (Cs) as positive control and comparison of Ar and Sc treatments (µg/mL).

**Figure 2 molecules-25-00476-f002:**
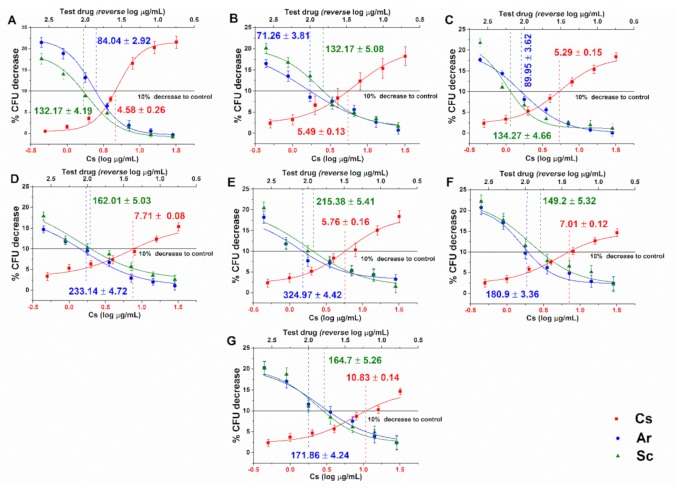
Minimum effective concentrations (MEC_10_) (mean ± SD) of artemisinin (Ar) and scopoletin (Sc) on *C. albicans* (**A**), *C. dubliniensis* (**B**), *C. tropicalis* (**C**), *C. krusei* (**D**), *C. glabrata* (**E**), *C. guilliermondii* (**F**) and *C. parapsilosis* (**G**) species. Six independent experiments, each in three replicates, compared with caspofungin (Cs) as positive control and comparison of Ar and Sc treatments (µg/mL).

**Figure 3 molecules-25-00476-f003:**
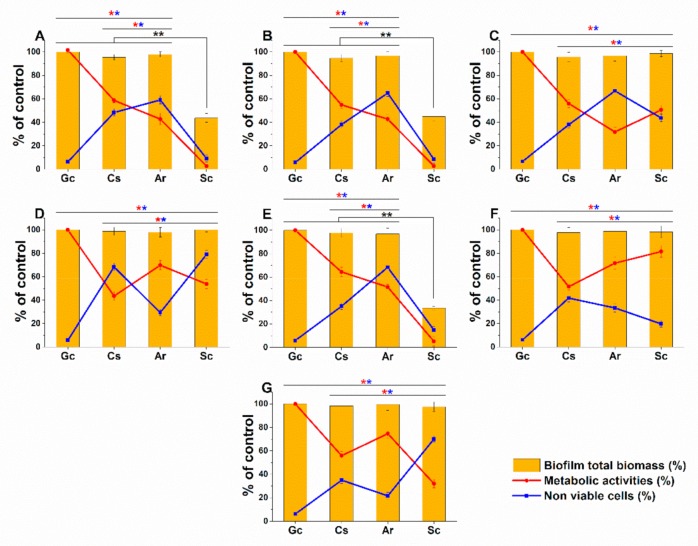
Effects of artemisinin (Ar) and scopoletin (Sc) at their MEC_10_ concentrations on the metabolic activity, amount of biofilm biomass, and viability of *C. albicans* (**A**), *C. dubliniensis* (**B**), *C. tropicalis* (**C**), *C. krusei* (**D**), *C. glabrata* (**E**), *C. guilliermondii* (**F**), and *C. parapsilosis* (**G**) cell populations (mean ± SD, n = 6 independent experiments, data were compared with untreated controls (Gc) and with caspofungin (Cs)-treated positive controls. The red (*****) and blue (*****) asterisks represent a significance value of *p < 0.01* for the metabolic activity and viability measurements, respectively. Whereas, the black double asterisks (******) highlight the changes in the Sc-treated biofilm biomass when compared to Gc, Cs, and Ar treatments at *p < 0.01* significance level.

**Figure 4 molecules-25-00476-f004:**
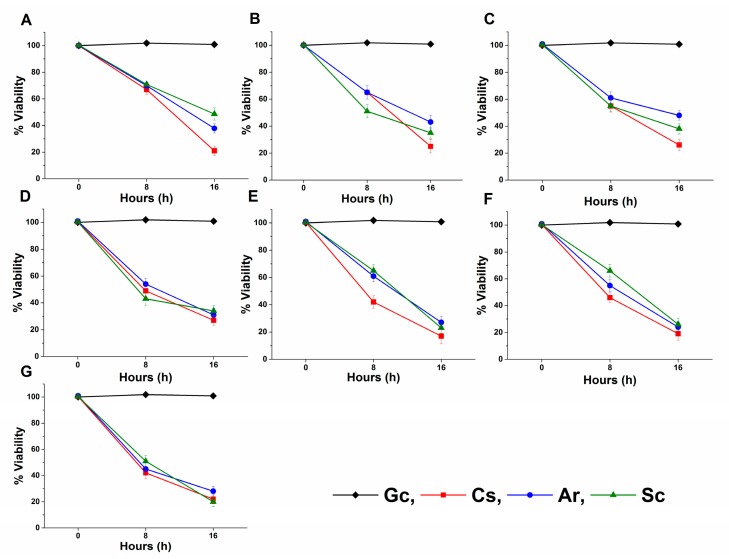
Effects of artemisinin (Ar) and scopoletin (Sc) at their MEC_10_ concentrations on the viability of planktonic *C. albicans* (**A**), *C. dubliniensis* (**B**), *C. tropicalis* (**C**), *C. krusei* (**D**), *C. glabrata* (**E**), *C. guilliermondii* (**F**), and *C. parapsilosis* (**G**) species compared with untreated controls (Gc) after 8 and 16 h of treatment (mean ± SD, n = 6 independent experiments, caspofungin (Cs) was used as positive control).

**Figure 5 molecules-25-00476-f005:**
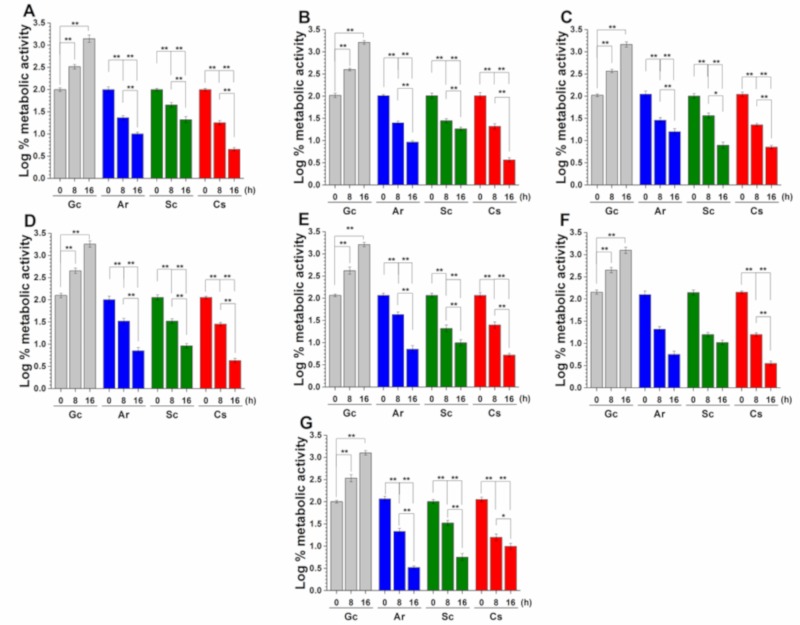
Effects of artemisinin (Ar) and scopoletin (Sc) at their MEC_10_ concentrations on the metabolic activities of planktonic *C. albicans* (**A**), *C. dubliniensis* (**B**), *C. tropicalis* (**C**), *C. krusei* (**D**), *C. glabrata* (**E**), *C. guilliermondii* (**F**), and *C. parapsilosis* (**G**) species compared with untreated controls (Gc) and caspofungin (Cs) as positive control after 8 and 16 h of treatment (mean ± SD, n = 6 independent experiments, **p < 0.05* and ***p < 0.01*).

**Figure 6 molecules-25-00476-f006:**
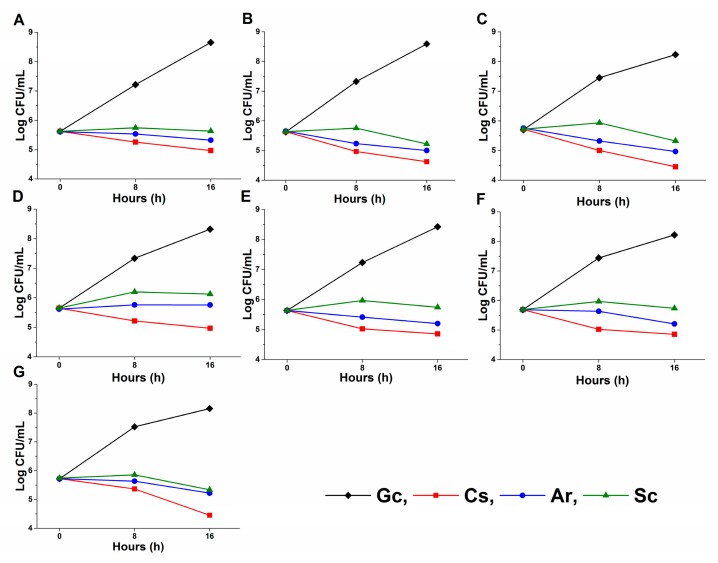
Effects of artemisinin (Ar) and scopoletin (Sc) at their MEC_10_ concentrations on colony formation of planktonic *C. albicans* (**A**), *C. dubliniensis* (**B**), *C. tropicalis* (**C**), *C. krusei* (**D**), *C. glabrata* (**E**), *C. guilliermondii* (**F**) and *C. parapsilosis* (**G**) species compared with untreated controls (Gc) after 8 and 16 h of treatment (mean ± SD, n = 6 independent experiments, caspofungin (Cs) was used as positive control).

**Figure 7 molecules-25-00476-f007:**
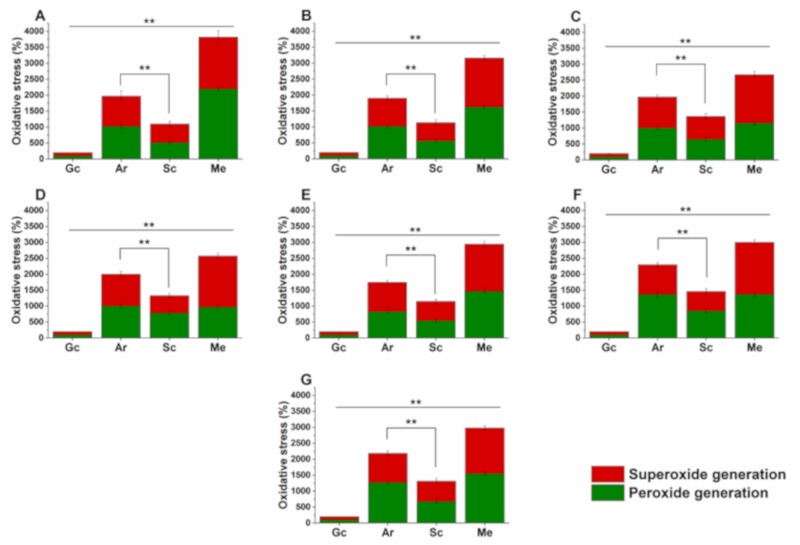
Effects of artemisinin (Ar) and scopoletin (Sc) at their MEC_10_ concentrations on peroxides (O_2_^2-^) and superoxide anion (O_2_^•-^) generation in planktonic *C. albicans* (**A**), *C. dubliniensis* (**B**), *C. tropicalis* (**C**), *C. krusei* (**D**), *C. glabrata* (**E**), *C. guilliermondii* (**F**), and *C. parapsilosis* (**G**) species compared with untreated (Gc), with menadione (Me)-treated controls and comparison of Ar and Sc treatments (mean ± SD, n = 6 independent experiments, ***p < 0.01*).

**Table 1 molecules-25-00476-t001:** *Candida* species examined in the study.

Species	Collection Code	Origin
*C. albicans*	SZMC 1372	clinical sample/Debrecen, Hungary
*C. dubliniensis*	CBS 7987 SZMC 1470	oral cavity of HIV-infected patient/Melbourne, Australia
*C. tropicalis*	SZMC 1368	clinical sample/Debrecen, Hungary
*C. krusei*	SZMC 0779	clinical sample/National Institute of Environmental Health (NIEH), Hungary
*C. glabrata*	SZMC 1374	clinical sample/Debrecen, Hungary
*C. guilliermondii*	SZMC 0808	clinical sample/Pécs, Hungary
*C. parapsilosis*	SZMC 8006	clinical sample/Szeged, Hungary

Abbreviations: SZMC, Szeged Microbiological Collection, Szeged, Hungary (http://www.wfcc.info/ccinfo/collection/by_id/987); CBS, Westerdijk Fungal Biodiversity Institute, Utrecht, The Netherlands (http://www.westerdijkinstitute.nl/).

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
