# Peer review of "Cytotoxic Action of Artemisinin and Scopoletin on Planktonic Forms and on Biofilms of Candida Species"

_molecules, 2020, doi:10.3390/molecules25030476_

Round 1

Reviewer 1 Report

The manuscript entitled "Cytotoxic action and anti-biofilm formation activities of artemisinin and scopoletin on Candida species" by Sourav Das et al., describes cytotoxic and anti-Biofilm activity of two well known anti-malarial plant metabolites against selected Candida spp.

The manuscript has been improved a lot after revision, Although, I have some minor comments to be corrected before accepted for publication;

1) In Abstract, (Line 22) authors already abbreviated artemisinin (Ar) and scopoletin (Sc), eventhough, they repeatedly used the full names in the abstract.
3) Line 45; “C. albicans is often found 45 in vaginal, mucosal and deep tissue infections [3]” should be omitted as it doesn’t go with the flow of introduction or can be added at later stage such as at line 51.
4) Section 2.2; Authors are requested to provide MEC10 values in digit also in the text.

5) Another very interesting point needed to be addressed by author is to correlate the difference between anti-biofilm activity and Inhibitory concentration and it should be added in the discussion section.

6) All the instrument details can be added in section 4.1 instead of repeating it throughout in section 4.

7) Why authors selected MEC10 concentrations for antibiofilm activity? Why not non-inhibitory concentration? All the compounds with little inhibition will affect its biofilm ability, but in order to get an effective anti-biofilm compound, it’s important to check activity at non-inhibitory concentration.

In this case, again it depends on the principle behind the assay, do author just want to check anti-biofilm activity or wants to show the anti-biofilm activity in terms of anti-virulence activity of the compound.

Reviewer 2 Report

My last comments were misunderstood.  1) The data in this manuscript doesn't show the inhibition of biofilm formation by Ar and Sc. The title tells two conclusions in this manuscript. The compounds have cytotoxic action on Candida species, and the compounds inhibit their biofilm formation. I agree with the first one, the cytotoxicity of the compounds were demonstrated in the manuscript.  However, inhibition of biofilm formation was not clearly shown in the manuscript. When the proliferation of Candida species are inhibited, mass of biofilm formation and metabolic activity will surely be decreased. For me, it looks like, the reduction of mass biofilm and metabolic activity correlate with the reduction of viable cells. If so, the compounds don't have specific activity to inhibit biofilm formation and the metabolic activity. Then, I agree with reviewer 1, the study lacks novelty to publish in Molecules.  If the inhibition is a conclusion of the manuscript, reduction of biofilm formation (and also metabolic activity) per viable cells must be compared and discussed.      2-1) "Abstract" tells what is described in main text. So, it shouldn't tell, anything isn't described in the main text. Again, I agree that antifungal activity of Ar and Sc were shown in the main text. But I don't find anything about the mode of action (fungistatic and/or fungicidal) in the main text. The "synergistic" idea (either with Cs or others) is not described in the main text, too. They must be described in the main text, or toned down in the abstract.    2-2) I'm not persuaded for line 230-231. Even if the compounds inhibit the biofilm formation, which data did demonstrate "the rapid penetration", and killing "the surface attached cells"? 
